# High pertussis circulation among infants, children and adolescents in Abidjan, Côte d'Ivoire

Man Koumba Soumahoro[1], Gaëlle Noel[2☉], Florence Campana[3☉], Constant Joseph Koné[1], Mahdi Rajabizadeh[3], Daouda Sévédé[4], Mohand Ait Ahmed[5], Kan Stéphane Kouassi[6], Yves Landry Kangah[1], Mariam Deme-Dramé[7], Kouamé Mathias N'Dri[7], Marie-Pierre Ouattara-Abina[8], Narcisse Tano[1], Fadima Sylla[9], Melissa Cardenat[10], Denis Macina[11], Nicole Guiso[12], Fabien Taieb[3]*

1 Epidemiology Unit, Institut Pasteur de Côte d'Ivoire, Abidjan, Côte d'Ivoire, 2 Center of Translational Research, Institut Pasteur, Paris, France, 3 Institut Pasteur Medical Center, Paris, France, 4 Unité de Sérologie Bactérienne et Virale, Institut Pasteur de Côte d'Ivoire, Abidjan, Côte d'Ivoire, 5 Clinical Research Coordination Center, Institut Pasteur, Paris, France, 6 Molecular Biology Platform, Institut Pasteur de Côte d'Ivoire, Abidjan, Côte d'Ivoire, 7 Pediatrics Ward, Hôpital Général de Port-Bouet, Abidjan, Côte d'Ivoire, 8 Pediatrics Ward for Children, Hôpital Général de Yopougon-Attié, Abidjan, Côte d'Ivoire, 9 Pediatrics Ward, Centre de Santé Urbain à Base Communautaire d'Angré, Abidjan, Côte d'Ivoire, 10 Pediatrics Ward, Centre Hospitalier et Universitaire d'Angré, Abidjan, Côte d'Ivoire, 11 Global Medical, Sanofi, Lyon, France, 12 Institut Pasteur, Paris, France

☉ These authors contributed equally to this work.
* fabien.taieb@pasteur.fr

**Data Availability Statement:** Preprocessed data will be shared via Zenodo. The data underlying results presented in the study will be available with the DOI 10.5281/zenodo.12911472.

## Abstract

### Background

Whooping cough due to *Bordetella pertussis* (BP) and/or *B. parapertussis* (BPP) is a highly contagious infection sometimes fatal for infants. Africa accounts for the largest share of cases and deaths worldwide. To evaluate pertussis circulation, we conducted a hospital-based prospective study (PS) including household contact-cases investigation (CCS) and a sero-epidemiological cross-sectional study (SECS).

### Methods

The PS, in which *Bordetella* diagnostics (qPCR) were performed, included infants aged ≤6 months presenting with ≥5 days of cough associated with one pertussis-like symptom. CCS was performed using qPCR and serology regardless of clinical signs. In the SECS, serology was performed in children aged 3–15 years with primary pertussis immunization.

### Results

Study took place in Abidjan between April 2019 and July 2021. In the PS, 187 infants with median age of 2.8 months were enrolled. A total of 42 (22.5%) were confirmed as positive, including 29 (15.5%), 4 (2.1%) and 9 (4.8%) of BP, BPP and BP/BPP coinfections respectively. Age <3 months, low BMI, apnea and inclusion period were identified as risk factors of infection. In the CCIS, 36 households were investigated, representing 158 people with

**Funding:** Initials: D.M. Grant Number: PER00061 Funder: Sanofi website: www.sanofi.com The sponsor/funder did not do any of these tasks.

**Competing interests:** Denis Macina is an employee of Sanofi and declares holding shares of the Sanofi group as part of his compensation. There are no patents, products in development or marketed products associated with this research to declare. This does not alter our adherence to PLOS ONE policies on sharing data and materials.

median age of 21.4 years. BP and/or BPP infection was confirmed for 77/157 (49.0%). Twenty-nine households (81%) had at least one positive case. Sixty-seven (42.7%) contact cases were categorized as possibly contaminated, mainly siblings older than five years (25.4%). Twenty-seven contact cases were considered as possible contaminators, primarily mothers (26%). In the SECS, 444 people were enrolled. Two hundred and thirty-eight (53.6%) and nine (2.0%) children had received one and two boosters, respectively. A positive serology was observed in 96 (21.7%) of children and adolescents.

## Conclusion

Our study highlighted high circulation of pertussis. Barriers to accessing boosters recommended by WHO need to be overcome. Laboratory capacities are key at individual level and to build an effective surveillance system.

## Introduction

Whooping cough is a highly contagious airborne infection which is especially severe in infants [1]. According to a modelling study based on United Nations population estimates and WHO and United Nations International Children's Emergency Fund (UNICEF) data, 160,700 children under five died in 2014 from *Bordetella pertussis* [2], the bacterium responsible for the disease. The same study found that Africa accounts for the largest proportion of pertussis cases (33%) and deaths (58%) occurring worldwide. However, these alarming numbers are unreliable and underestimated [3–6] since Africa is made up of many low- and middle-income countries (LMIC) that lack reliable surveillance and diagnostic laboratory capacity.

Whooping cough is a vaccine-preventable disease and vaccination in LMIC, through the Expanded Program on Immunization (EPI) launched by WHO in 1974 with a pertussis whole-cell vaccine (wP) composed of inactivated bacterial suspensions, has greatly reduced the global disease burden [1]. However, the lack of reliable surveillance in sub-Saharan Africa hinders the optimal use of resources and development of vaccination strategies adapted to the local disease epidemiology [5]. What data are needed? First and foremost, *B. pertussis* circulation needs to be accurately assessed in order to identify the true disease burden and inform decision-makers. Secondly, vaccination coverage (the proportion of eligible, vaccinated individuals) and compliance need to be reliably estimated since high infection rates can be caused by nonadherence to recommendations. Finally, wP efficacy and effectiveness can vary greatly across producers [7], as shown in the 1990s [8] and again recently [9, 10]. The WHO EPI schedule consists of three-dose primary vaccination at six, ten and fourteen weeks of age in most sub-Saharan African countries including Côte d'Ivoire [9, 11, 12], where the national program also recommends vaccine boosters at 16 months and then two additional boosters at five-year intervals at 6.5 or DTCP and 11.5 years [13]. The vaccines are injected at 6, 10 and 14 weeks in Côte d'Ivoire are wP vaccines (DTC-HepB-Hib, e.g. by Serum Institute of India), reflecting the contrast with high-income countries, where acellular vaccines, that are less reactogenic than wP [1, 14], are used for primary vaccination. Noteworthy, all boosters in Côte d'Ivoire are acellular vaccines (e.g. DTCP (TETRAXIM®)); DTCP (Adacel®) and DTP (Dultavax®)), and are out-of-pocket expenses. High vaccine coverage and compliance with this three-dose schedule for infants in their first year of life is not sufficient for *B. pertussis* to be controlled. It has been known since the late 1980s in the US and since the late 1990s in Europe

that primary vaccination and one booster with wP, like natural infection, do not provide life-long protection and vaccine boosters are needed [15, 16]. The waning of infection- and vaccine-induced protection has shifted the infection towards older children, adolescents and adults, who have been shown to be a source of infection among unvaccinated (or not fully vaccinated) infants [17]. The infection shift toward older individuals has recently been confirmed in the Maghreb, South Africa, Senegal, Gambia, Togo, Madagascar and Niger [5, 18]. Therefore, local changes in pertussis epidemiology in sub-Saharan Africa must be assessed to inform policies that could limit these age groups' roles in infant infections.

Building on the work done in Togo, Madagascar [18], Cambodia [19] and Iran [20, 21], we conducted a study in Côte d'Ivoire, in Abidjan. Our general goal was to establish a central laboratory with sensitive and specific biological diagnosis to better understand pertussis circulation in Abidjan, which is not documented to our knowledge, and to assess *B. pertussis*, *B. parapertussis* and *B. holmesii* circulation among infants as well as their clinical presentation and short-term evolution.

To reach this goal, we set up a PCR and serology laboratory at the Institut Pasteur de Côte d'Ivoire and initiated a study in two parts. The first part consisted of a hospital-based prospective study (PS) of infants under six months of age and presenting with pertussis-like syndrome and included contact cases investigation study within the households of identified index cases. The second part consisted of a sero-epidemiological cross-sectional study among children and adolescents.

## Methods

### Study population and design

**Hospital-based prospective study and household contact-cases investigation.** Infants were recruited from April 2019 to July 2021 in four hospitals in various Abidjan districts, for a prospective study in which infants were followed up for an assessment of the evolution of the disease. Male and female infants were eligible for enrollment if they were aged ≤6 months with a persistent cough for at least five days associated with at least one symptom among apnea, inspiratory "whooping" or post-coughing vomiting, or with a confirmed case of whooping cough in the entourage. After inclusion, one nasopharyngeal swab (NPS) in each nostril was collected for PCR analysis. Data on age, sex, size and weight, birth information (weight and term at birth), pertussis vaccination history based on vaccination booklets, type and duration of pertussis-related symptoms, biological test results (if available), antibiotic treatment type and duration, previous consultations, family composition and cases of cough in the entourage were recorded. Pertussis-related symptoms, antibiotic treatment, hospitalization and disease outcome were followed up fourteen days after inclusion.

For each positive (index) case, contact-case investigation study was proposed to any person in close contact with the index case in his/her household, whether they lived in it or not. The inclusion criterion for close contacts was regular contact (>1 h/day) with the index case for at least five days before symptom onset in the infant. Upon enrollment of contact cases, one NPS in each nostril and one capillary blood sample were collected. Data on age, gender, the relationship with the index case, pertussis vaccination history based on vaccination documents, type and duration of past and current pertussis-related symptoms, antibiotic treatment type and duration, and previous consultations were recorded.

Data were collected on standardized paper questionnaires and recorded in a computerized database by two different data clerks using a double-entry method.

**Sero-epidemiological cross-sectional study.** Children and adolescents were recruited from January to June 2021 in fifteen schools. Male and female children and adolescents were

eligible for enrollment if they were aged 3–15 and had received a complete primary pertussis vaccination (three doses), with the last dose at least one year earlier than the sampling date. Presenting proof of vaccination (i.e. a vaccination booklet or record) was necessary to be included. People with a known hemostasis disorder, serious acute or chronic illness, or any illness that kept the child from attending the appointment for blood collection were not included in the study. After inclusion, one fingerpick blood sample was taken for serology analysis. Information on pertussis vaccination history, sex, age and blood sampling was collected on a standardized questionnaire and recorded in a computer database by two different data clerks using a double-entry method. No information on past or current respiratory illness was collected.

## Biological sample collection and analysis

To ensure the quality and accuracy of results, a double validation was conducted and, when the result interpretation was inconsistent between the Institut Pasteur and Institut Pasteur de Côte d'Ivoire, the test was repeated by Institut Pasteur Côte d'Ivoire. After test repetition, all the samples were interpretable and could thus be included in the analysis.

**Real-time quantitative Polymerase Chain Reaction (qPCR).** Quality control samples and *B. pertussis*, *B. parapertussis* and *B. holmesii* positive control samples were provided by Quality Control for Molecular Diagnostics (QCMD, Glasgow Scotland), an independent International External Quality Assessment organization. Quality controls were blind tested before the study began.

As previously described [20], samples were collected by nurses in a standardized manner across the study sites. NPS samples were taken using eSwab-containing Amies transport medium (Dacron, Ref. 482CE). Samples were sent to the Molecular Biology Platform at the Institut Pasteur de Côte d'Ivoire.

NPS were kept at 4˚C before DNA extraction within 48 hours. When not possible, the sample was stored at -80˚C until processing, a temperature which has been shown to preserve DNA integrity [22]. Samples were homogenized by vortex and the swab was discarded. A dilution of the extraction was realized: first, total DNA was extracted from 100 μl of the transport media (pure sample). Then, 10 μl of the transport media were spiked into 90 μl of sterile water to obtain a diluted sample using the commercial High Pure PCR Template Preparation Kit by Roche [23]. For each sample, both pure and diluted DNA extract solutions were tested by qPCR: the diluted DNA extract solution was meant to detect PCR inhibition in the pure sample caused by too large quantity of DNA, and thus to avoid false negative qPCR results. qPCR was carried out using the commercial LightCycler 480 Probes Master kit by Roche [24], 0.5 μM forward and reverse primers and 0.2 μM TaqMan probe by TIB Molbiol [25], 2.5% DMSO by Sigma [26] and 5 μl of DNA extract solution. The amplification of insertion elements IS481 (*Bordetella spp.*) and IS1001 (*B. parapertussis*) was first assessed. IS481 qPCR is very sensitive but not specific as it targets the DNA of both *B. pertussis* and *B. holmesii* [21]. Therefore, IS481+ samples were tested for ptxP, IS1002 and hIS1001 amplification to identify *B. pertussis* (ptxP+ and/or IS1002+) and *B. holmesii* (hIS1001+). In cases of IS481+ samples with neither IS1002 nor hIS1001 amplification, we considered the sample as "Bor+" (S1 Fig). qPCR analyses were carried out using the LightCycler 96 Instrument (Roche) and, after validation, CFX96 (Bio-Rad). The sequence of RNase P human gene was amplified to ascertain the quality of sampling and laboratory procedures. Purified DNA from *Bordetella* reference strains and non-template control samples (PCR-grade water) were included as positive and negative qPCR control samples, respectively, in each qPCR.

**Serology analysis.** One blood sample (200–400 μl) was also collected in SST microtainer tubes from a fingertip using a 23G lancet needle (Becton Dickinson, ref. 365968 and 369523, respectively). Once received by the Bacterial and Viral Serology Unit (USBV) at Institut Pasteur of Côte d'Ivoire, the blood was immediately spun and the serum was stored at -20˚C for further analysis.

As previously described [18–20], anti-PT IgG titers were quantified using a commercial purified PT-containing enzyme-linked immunosorbent assay (ELISA) kit (EUROIMMUN; reference EI 2050-G) [27] and the WHO reference serum available from the National Institute for Biological Standards and Control (NIBSC, newly MHRA). All tests had internal negative and positive controls and passed the validity criteria. The results were reported as International Units (IU)/ml. The lower limit of quantitation (LLOQ) was defined as 5 IU/ml.

## Case definition

**Definition of *B. pertussis* and *B. parapertussis* cases (hospital-based prospective and contact-case investigation studies).** Full-range cycle threshold (CT) value was considered for IS*481*+ qPCR. Among IS*481*+ samples, *B. pertussis* infection was molecularly confirmed with IS*1002*+ and/or *ptxP*+, and/or epidemiologically confirmed when, for an individual identified as a BOR+ case, a *B. pertussis* species was identified in at least one contact case by serology or qPCR. *B. parapertussis* infection was molecularly confirmed with IS1001+. *B. pertussis* infection could not be confirmed for all IS*481*+ biological samples. These individuals were considered as negative cases. Therefore, cases were considered positive in cases of *B. pertussis* and/or *B. parapertussis* infection. BOR+ cases were considered negative in the main analysis.

**Potential contaminators/contaminated among contact cases.** In the contact-case investigation study, people with positive PCR and/or serology were classified as potential contaminators or as potentially contaminated. This classification was based on serology and PCR results, the onset of coughing among contact cases relative to their index case, clinical symptoms presented at the moment of or before inclusion and vaccination history. We did not consider serology results for participants with less than one year after their last vaccine injection or for households linked to an index case infected by *B. parapertussis* only.

**Recent history of pertussis infection (sero-epidemiological study).** A titer of <5 IU/ml indicated seronegative status and children with anti-PT IgG levels ≥5 IU/ml were considered seropositive. Anti-PT titers were categorized using 40 IU/mL and 100 IU/ml cut-offs to evidence children who had had contact with *B. pertussis* sometime during the past 12 months when 40–100 IU/ml and 3 months when ≥100 IU/mL [27].

Since all the participants had received their last dose at least one year before inclusion, serology results were due to contact with the bacterium and not induced by vaccine injections.

Analysis of serology results according to the age of evaluated children was restricted to a population vaccinated on time, defined as follows: i) for all the children, aged under six months at the time of the primary vaccination's third-dose injection; ii) for children who had received one booster: aged 12 to 24 months at the time of the first booster injection and more than a six-month interval between the primary vaccination's third-dose injection and the first booster injection.

## Statistical analyses

For continuous variables, medians, interquartile ranges [IQR], minimums and maximums were calculated. Comparisons were made using Chi-square tests, and Fischer exact test whenever the number of observations in any category was below five, and the non-parametric Wilcoxon Rank Sum test was used for continuous variables. When necessary, continuous

variables were categorized based on median values or on values commonly acknowledged in the literature. Spearman's coefficient was used to assess rank correlation. For categorical variables, proportions were computed and comparisons made using the two-sided Chi-square or Fisher's exact test.

In the PS study, anthropometric z-scores for weight-for-height, height-for-age and Body Mass Index (BMI) for age were calculated and interpreted using the WHO and Centers for Disease Control and Prevention Child Growth Standards [28]. Weight-for-height, height-for-age and BMI were interpreted as abnormally low when the corresponding z-score was more than two standard deviations below the WHO Child Growth Standards median. The scores were computed using the zscorer package in R version 4.3.1. Data on infants' weight and term at birth were obtained from their health booklets. Data from infant hematological examinations were considered when generated up to three days before inclusion.

To define the timeliness of pertussis immunization, and due to the population of patients recruited (aged ≤6 months) in the PS study, we assessed compliance as the number of doses received by an individual relative to the number of doses expected at their age among eligible children (i.e. older than 6 weeks). Following the pertussis vaccination schedule in Côte d'Ivoire, timeliness with the national schedule was defined as having received the first injection at ≥38 days and before 10 weeks of life, and the second and third injections 17 days to 5 weeks after the previous dose. For the sero-epidemiological study, we presented compliance using the stringent definition presented above and compliance using a previously published, less stringent definition focused on immune response generated by primovaccination [18–20]. The latter was based on intervals between doses rather than age, where respecting intervals of 24 days to 10 weeks fulfills the compliance criterion, with no upper age limit for the first dose.

To evaluate the risk factors associated with whooping cough infection (identified through a positive PCR), the factors associated with the outcome with a p-value ≤0.20 in the univariate analysis were considered in the multivariate analysis by logistic regression. A sensitivity analysis was first performed by removing BOR+ cases and secondly by considering them as positive cases.

Statistical analyses were performed using Stata 18.0 (StataCorp LLC, Texas, USA).

## Ethics approval and consent to participate

The study was sponsored by the Institut Pasteur, Paris. The study protocol was reviewed and approved by the Institut Pasteur's Institutional Review Board (IRB00006966) and by the Ministry of Health's National Research Ethics Committee in Côte d'Ivoire (057-18/MSHP/CNER-kp). The procedures followed were in accordance with the World Medical Association's Declaration of Helsinki (2008). Informed written consent was obtained from both adult participants and the parent(s)/legal guardian(s) of all under 18s (minor children), and oral assent was obtained from children aged over 7 years. Only individuals who agreed to participate were included in the study. Authorization for data processing was obtained from the French Data Protection Authority (CNIL), and the data was pseudonymized with a study-specific code for each participant. The study's ClinicalTrials.gov identifier is NCT03388034.

## Results

### Hospital-based prospective study

**Study population.** A total of 187 infants were enrolled over a period of 28 months (Fig 1).

Among the enrolled participants, the gender ratio (M/F) was 0.99 and the median [IQR] age was 2.8 [1.8–4.6] months, ranging from 12 days to 6 months. The population description is shown in Table 1. Briefly, 94 (50.3%) infants received two or three vaccine injections against

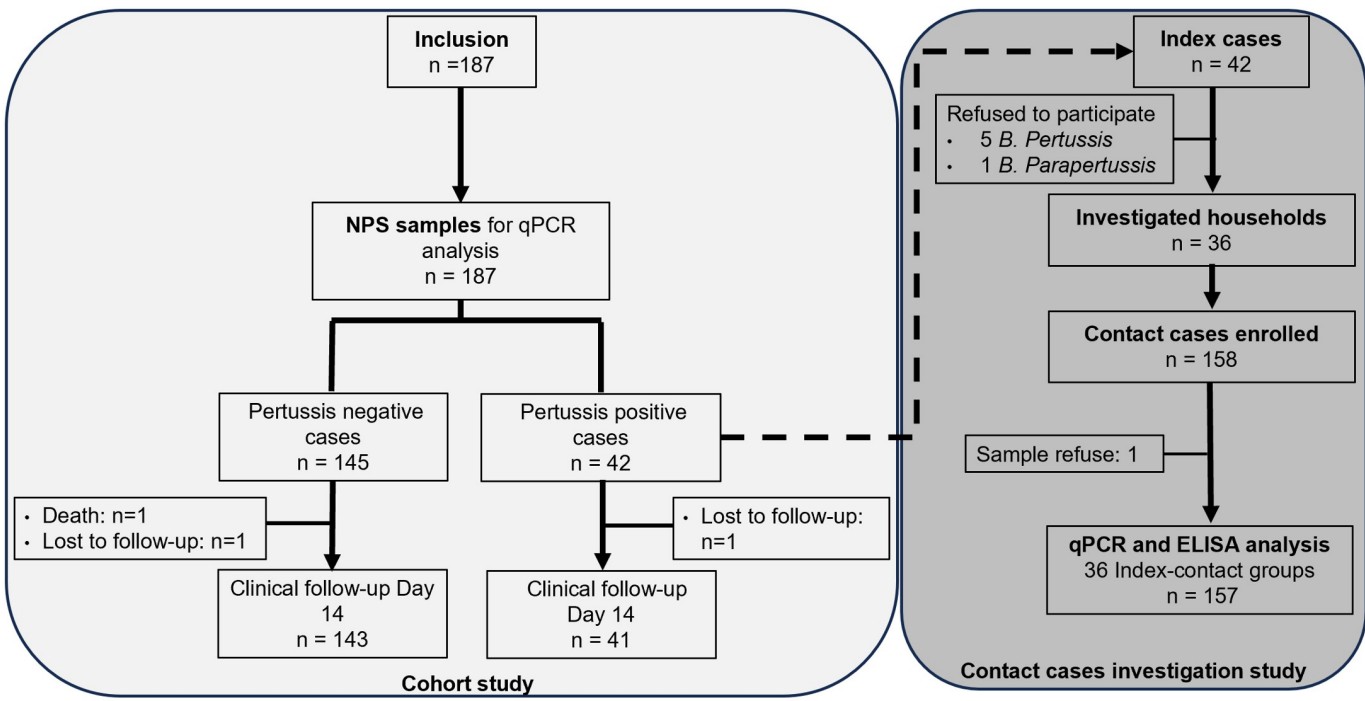

**Fig 1. Flowchart of the hospital-based prospective study and contact cases investigation study.** NPS: Nasopharyngeal samples.

pertussis. Considering age, 123 of the 164 (75.0%) infants eligible for vaccination by virtue of their age were compliant with the EPI. Thirty-one (16.6%) patients presented with a low birth weight. Before inclusion, 108 (57.8%) infants had previously attended at least one medical consultation and 79 (42.2%) had been exposed to antibiotics. Among these 79 infants, nine (4.8%) had received macrolides, six of them more than three days before inclusion (three were exposed to Josamycin and three to Azithromycin). The median [IQR] duration of cough was 8 [7–14] days. Five infants presented with a cough for more than 45 days. Regarding clinical symptoms, apnea, cyanosis, inspiratory whoop and post-tussive vomiting were declared before inclusion for 30 (16.0%), 4 (2.1%), 4 (2.1%) and 56 (29.9%) patients respectively.

At the time of inclusion, 33 (17.7%) infants had a Body Mass Index (BMI) for age lower than 2 standard deviations. Apnea, cyanosis, inspiratory whoop and post-tussive vomiting were observed for 55 (29.4%), 3 (1.6%), 7 (3.7%) and 137 (73.3%) patients respectively. Fever was noted for 38 (20.3%) patients and the median [IQR] oxygen saturation was 94% [89–98]. An abnormal pulmonary auscultation was observed for 113 (60.4%) infants.

Regarding household characteristics, the median [IQR] number of people per household was 6 [4–8], including 3 [2–4] and 3 [2–4] adults and children respectively. Cough in the entourage was reported for 91 (48.7%) households.

Biological analyses were performed for 57 infants (S1 Table).

*Diagnosis of pertussis and parapertussis infections.* Pertussis and/or *parapertussis* infection testing was provided for the 187 infants included. A total of 42/187 (22.5%) infants were confirmed as positive cases. There were 29 (15.5%) infected with *B. pertussis* only, 9 (4.8%) coinfected with *B. parapertussis* and 4 (2.1%) infected with *B. parapertussis* only. Therefore, a total of 38 infants infected with *B. pertussis*, including coinfections, were observed. Among negative infants, one was confirmed to have a BOR+ infection, but epidemiological confirmation was

**Table 1. Population description.**

| Population | Total n = 187 | Positive cases n = 42 | Negative cases n = 145 | p-value |
|---|---|---|---|---|
| Gender, n (%) | | | | 0.15 |
| *Girls* | 94 (50.3) | 17 (40.5) | 77 (52.4) | |
| Age (months) | | | | |
| *Median [IQR]* | 2.8 [1.8–4.6] | 2.5 [1.6–3.6] | 3.2 [1.9–4.7] | 0.09[†] |
| *< 3 months, n (%)* | 98 (52.4) | 29 (69.1) | 69 (47.6) | **0.01** |
| *≥ 3 months, n (%)* | 89 (47.6) | 13 (30.9) | 76 (52.4) | |
| **Vaccination** | | | | |
| DTPcv* injection | | | | **0.03** |
| *0–1* | 92 (49.2) | 27 (64.3) | 65 (44.8) | |
| *2–3* | 94 (50.3) | 15 (35.7) | 79 (54.5) | |
| *missing values* | 1 (0.5) | 0 (0) | 1 (0.7) | |
| EPI compliance (age ≥6 weeks) | n = 164 | n = 34 | n = 130 | 0.88 |
| *Yes* | 123 (75.0) | 26 (76.5) | 97 (74.6) | |
| *missing values* | 1(0.6) | 0 (0) | 1 (0.8) | |
| DTPcv* injection (age ≥6 weeks) | | | | |
| *0–1* | 69 (42.1) | 19 (55.9) | 50 (38.5) | 0.07 |
| *2–3* | 94 (57.3) | 15 (44.1) | 79 (60.8) | |
| *missing values* | 1 (0.6) | 0 (0) | 1 (0.8) | |
| **Infant's morphological characteristics** | | | | |
| Low birth weight, n (%) | | | | 0.63 |
| *Yes* | 31 (16.6) | 6 (14.3) | 25 (17.2) | |
| *missing values* | 2 (1.1) | 0 (0) | 2 (1.4) | |
| Low weight-for-height, n (%) | | | | 0.17 |
| *Yes* | 26 (13.9) | 8 (19.1) | 18 (12.4) | |
| *missing values* | 31 (16.6) | 10 (23.8) | 21 (14.5) | |
| Low height-for-age, n (%) | | | | **0.01** |
| *Yes* | 48 (25.6) | 16 (38.1) | 32 (22.1) | |
| *missing values* | 28 (15.0) | 9 (21.4) | 19 (13.1) | |
| Low BMI for age, n (%) | | | | 0.13 |
| *Yes* | 33 (17.6) | 10 (23.8) | 23 (15.9) | |
| *missing values* | 28 (15.0) | 9 (21.4) | 19 (13.1) | |
| **Antibiotic therapy before inclusion** | | | | |
| Antibiotic therapy, n (%) | | | | **0.02** |
| *Yes* | 79 (42.3) | 25 (59.5) | 54 (37.2) | |
| *Macrolide exposure, n (%)* | 9 (11.4) | 5 (20.0) | 4 (7.4) | 0.13 |
| *missing values* | 6 (3.2) | 0 (0) | 6 (4.1) | |
| **Duration of cough at inclusion** | | | | |
| *median (days) [IQR]* | 8 [7–14] | 9 [7–21] | 8 [6–14] | 0.09[†] |
| *min–max* | 5–101 | 6–45 | 5–101 | |
| Duration of cough ≥7 days, n (%) | 142 (75.9) | 36 (85.7) | 106 (73.1) | 0.09 |
| Duration of cough ≥14 days, n (%) | 54 (28.9) | 16 (38.1) | 38 (26.2) | 0.13 |
| **Symptoms declared before inclusion** | | | | |
| Nocturnal cough, n (%) | 90 (48.1) | 21 (50.0) | 69 (47.6) | 0.76 |
| Coughing spells, n (%) | 103 (55.0) | 23 (54.8) | 80 (55.1) | >0.9 |
| Apnea, n (%) | 30 (16.0) | 12 (28.6) | 18 (12.4) | 0.01 |
| Cyanosis, n (%) | 4 (2.1) | 2 (4.8) | 2 (1.4) | 0.21 |
| Inspiratory whoop, n (%) | 4 (2.1) | 1 (2.4) | 3 (2.07) | >0.9 |
| Difficulty breathing, n (%) | 62 (33.1) | 13 (30.9) | 49 (33.8) | 0.72 |

*(Continued)*

**Table 1.** (Continued)

| Population | Total n = 187 | Positive cases n = 42 | Negative cases n = 145 | p-value |
|---|---|---|---|---|
| Vomiting, n (%) | 63 (33.6) | 16 (38.0) | 47 (32.4) | 0.35 |
| Post-tussive vomiting, n (%) | 56 (29.9) | 15 (35.7) | 41 (28.3) | 0.24 |
| Good condition between cough, n (%) | 96 (51.3) | 22 (52.4) | 74 (51.0) | >0.9 |
| **Clinical assessment at inclusion** | | | | |
| Coughing spells, n (%) | 184 (98.4) | 42 (100) | 142 (97.9) | >0.9 |
| Apnea, n (%) | 55 (29.4) | 15 (35.7) | 40 (27.6) | 0.31 |
| Cyanosis, n (%) | 3 (1.6) | 1 (2.4) | 2 (1.4) | 0.53 |
| *missing values* | 1 (0.5) | 1 (2.4) | - | |
| Inspiratory whoop, n (%) | 7 (3.7) | 1 (2.4) | 6 (4.1) | >0.9 |
| Difficulty breathing, n (%) | 131 (70.1) | 28 (66.7) | 103 (71.0) | 0.59 |
| Vomiting, n (%) | 143 (76.5) | 34 (81.0) | 109 (75.2) | 0.44 |
| Post-tussive vomiting, n (%) | 137 (73.3) | 31 (73.8) | 106 (73.1) | 0.93 |
| Good condition between cough, n (%) | 163 (87.2) | 39 (92.9) | 124 (85.5) | 0.42 |
| *missing values* | 2 (1.0) | - | 2 (1.4) | |
| SpO2, median [IQR] | 94 [89–98] | 91 [79–99] | 96 [90–98] | 0.41 |
| *missing values* | 2 (1.0) | 1 (2.4) | 1 (0.7) | |
| Abnormal pulmonary auscultation, n (%) | 113 (60.4) | 21 (50.0) | 92 (63.5) | 0.12 |
| Fever, n (%) | 38 (20.3) | 7 (16.7) | 31 (21.4) | 0.50 |

*DTP-containing vaccines (Diphtheria, Tetanus and Pertussis);

[†]rank-sum test

not possible due to the lack of contact. This participant was considered a negative case. No *B. holmesii* carriage was detected.

The characteristics of positive and negative cases are presented in Table 1.

Among positive cases, 10 (23.8%) had received 3 vaccine doses, representing 18.9% of the 53 patients fully vaccinated as recommended by the EPI. Among them, six were infected by *B. pertussis* only, one by *B. parapertussis* only and three were coinfected. One patient had received the third dose after symptoms began and another had received the third dose only four days before symptom onset. The 8 other infected infants had received their last dose at least 19 days before symptom onset, with a median delay of 38.5 days (IQR [23–54]). The clinical presentation of infected infants having received less than two doses vs. at least two doses did not appear as significantly different (S5 Table). For this analysis, a vaccine dose was considered only if received at least 14 days before symptom onset.

Among negative cases, three were exposed to macrolides more than three days before inclusion (two with Azithromycin and one with Josamycin). Regarding hematological analysis, the level of lymphocytes tended to be higher among positive pertussis cases (p = 0.09) (S1 Table).

In the multivariate analysis, age under three months, a low BMI for age and apnea were independently and significantly associated with a higher risk of pertussis infection (p<0.01, p = 0.04 and p = 0.01, respectively) (Table 2). Infants recruited before June 2020 also presented with a higher risk of pertussis infection compared with infants recruited in or after July 2020.

By focusing on the 94 infants receiving 2 or 3 doses of vaccine injections, 15 of whom were infected, low BMI for age (aOR 3.9 [1.1–14.6], p = 0.04) and apnea (aOR 4.4 [1.2–15.8], p = 0.02) were significantly associated with a higher risk of pertussis infection.

Sensitivity analysis, either by removing the Bor+ case or considering it a positive case, identified the same risk factors of pertussis infection in the multivariate analysis.

**Table 2. Factors associated with pertussis infection (logistic regression, n = 187, goodness of fit by Hosmer-Lemeshow test: p = 0.66).**

| | Positive cases n(%) | OR | SE | CI(95%) (p-value) | ORa | SE | CI (95%) (p-value) |
|---|---|---|---|---|---|---|---|
| Age | | | | | | | |
| < 3 months (n = 98) | 29 (69.1%) | **2.5** | **0.9** | **[1.2–5.1] (0.02)** | **3.5** | **1.5** | **[1.6–7.9] (<0.01)** |
| ≥ 3 months (n = 89) | 13 (31.0%) | ref† | ref | - | ref | ref | - - |
| Sex | | | | | | | - - |
| Boys (n = 93) | 25 (59.5%) | ref | ref | - | - | | |
| Girls (n = 94) | 17 (40.5%) | 0.6 | 0.2 | [0.3–1.2] (0.15) | NS* | | |
| DTPCV** injection | | | | | | | - - |
| 0–1 (n = 92) | 27 (64.3%) | ref | ref | - | - | | |
| 2–3 (n = 94) | 15 (35.7%) | **0.5** | **0.2** | **[0.2–0.9] (0.03)** | NS | | |
| Number of people in household | | | | | | | - - |
| ≤ 2 (n = 75) | 15 (35.7%) | ref | ref | - | - | | |
| >2 (n = 112) | 27 (64.3%) | 1.6 | 0.6 | [0.8–3.2] (0.16) | NS | | |
| Antibiotherapy before inclusion | | | | | | | - - |
| No (n = 102) | 17 (40.5%) | ref | ref | - | - | | |
| Yes (n = 79) | 25 (59.5%) | **2.3** | **0.8** | **[1.1–4.7] (0.02)** | NS | | |
| Low BMI for age | | | | | | | |
| No (n = 126) | 23 (54.8%) | ref | ref | - | ref | ref | - |
| Yes (n = 33) | 10 (23.8%) | 1.9 | 0.9 | [0.8–4.6] (0.13) | **2.7** | **1.4** | **[1.1–7.3] (0.04)** |
| Duration of cough, n (%) | | | | | | | - - |
| ≤7 days (n = 45) | 6 (14.3%) | ref | ref | - | - | | |
| >7 days (n = 142) | 36 (85.7%) | 2.2 | 1.1 | [0.9–5.6] (0.10) | NS | | |
| Normal pulmonary auscultation, n (%) | | | | | | | - - |
| No (n = 113) | 21 (50.0%) | ref | ref | - - | - | | |
| Yes (n = 74) | 21 (50.0%) | 1.7 | 0.6 | [0.9–3.5] (0.12) | NS | | |
| Apnea declared before inclusion | | | | | | | |
| No (n = 157) | 30 (71.4%) | ref | ref | - | ref | ref | - |
| Yes (n = 30) | 12 (28.6%) | **2.8** | **1.2** | **[1.2–6.5] (0.01)** | **3.3** | **1.5** | **[1.3–8.2] (0.01)** |
| Cyanosis declared before inclusion | | | | | | | - - |
| No (n = 183) | 40 (95.2%) | ref | ref | - | - | | |
| Yes (n = 4) | 2 (4.8%) | 3.9 | 3.6 | [0.5–29.0] (0.21) | NS | | |
| Number of adults in household | | | | | | | - - |
| 1–2 (n = 75) | 15 (35.7%) | ref | ref | - | - | | |
| 3–4 (n = 66) | 13 (31.0%) | 1.0 | 0.4 | [0.4–2.2] (0.96) | NS | | |
| >4 (n = 46) | 14 (33.3%) | 1.8 | 0.8 | [0.8–4.1] (0.19) | NS | | |
| Inclusion period | | | | | | | |
| April 2019 to June 2020 (n = 110) | 31(73.8%) | **2.4** | **0.9** | **[1.1–5.0] (0.03)** | **2.8** | **1.2** | **[1.2–6.5] (0.02)** |
| July 2020 to July 2021 (n = 77) | 11(26.2%) | ref† | ref | - - | ref | ref | - - |

*NS: not significant

**DTP-containing vaccines (Diphtheria, Tetanus and Pertussis); †rank-sum test

† ref: reference used for comparisons

*Clinical status at D14.* The final evaluation was performed at a median timepoint of 13 days [IQR: 13–14] ranging from 13 to 21 days.

One hundred and sixty-eight (89.8%) infants fully recovered (Table 3). The proportion of infants with persistent cough at Day 14 was significantly higher among positive cases (p<0.01). One infant died (negative case) and two were lost to follow-up but known to be alive at the time of evaluation.

**Table 3. Clinical status at the end of follow-up.**

| Clinical status | Total n = 187 | Positive cases n = 42 | Negative cases n = 145 | p-value |
|---|---|---|---|---|
| Full recovered, n (%) | 168 (89.8) | 32 (76.2) | 136 (93.8) | **p<0.01** |
| Improvement with persistent cough, n (%) | 14 (7.5) | 8 (19.0) | 6 (4.1) | **p<0.01** |
| No improvement, n (%) | 2 (1.1) | 1 (2.4) | 1 (0.7) | - |
| Death, n (%) | 1 (0.5) | 0 (0) | 1 (0.7) | - |
| Lost to follow-up, n (%) | 2 (1.1) | 1 (2.4) | 1 (0.7) | - |

## Household contact-cases investigation

**Population study.** During the contact-case investigation, 36 households with positive cases were investigated. Among the households investigated, 24 were related to a *B. pertussis* index case, 3 to a *B. parapertussis* index case and 9 to a *B. pertussis* and *parapertussis* coinfection index case. Household contacts of six positive cases refused to take part in the study. A total of 158 individuals were recruited. One participant refused the nasopharyngeal and blood sampling (Fig 1).

The median [IQR] age of contact cases was 21.4 [6.2–35.1] years, ranging from 4 months to 72 years. Among them, 102 (64.6%) were females and 91 (57.6%), 28 (17.7%) and 39 (24.7%) were adults (i.e. older than 15 years), children aged 6 to 15 and children under 5, respectively. Regarding clinical symptoms, 77 (48.7%) participants presented with respiratory symptoms consistent with whooping cough (S2 Table).

The households investigated were composed of 5 [3–5] people included in median [IQR], ranging from 2 to 8. Contact cases consisted of 30 (19.0%) mothers; 29 (18.4%) aunts and uncles; 29 (18.4%) siblings and cousins older than 5 years; 18 (11.4%) siblings and cousins younger than 5 years; 17 (10.8%) individuals sleeping in the same bedroom without necessarily being family members; 13 (8.2%) fathers; 11 (7.0%) grandparents and great-grandparents and 11 (7.0%) nurses or cleaners (S2 Table).

**Diagnosis of pertussis and parapertussis infections.** The median [IQR] interval between index identification and household investigation was 9 [7–11] days and ranged from 5 to 19 days. The median [IQR] interval between the onset of symptoms in index cases and the contact case investigation was 17 [15–25] days, ranging from 12 to 40 days. Nearly half of contact cases (77 (48.7%)) presented with symptoms, mainly cough (S2 Table).

*B. pertussis* and/or *parapertussis* infection was confirmed by PCR and/or serology for 77/157 (49.0%) contact cases, among which 37/157 (23.4%) by PCR. *B. pertussis*, *parapertussis* infection and coinfection were found for 31, 3 and 3 individuals respectively. No *B. holmesii* carriage was detected. Regarding serology results, 59/157 (37.6%) people presented with a result above the 40 UI/mL threshold and, among them, 26 (16.5%) had a result above 100 UI/mL. The median [IQR] number of positive cases by PCR or serology per household was 3 [1–3], ranging from 0 to 6, with a median of 50% [18–80] positive cases among all the contacts in a household, ranging from 0 to 100%. A large majority of households (29/36, 81%) had at least one positive case. If only pertussis-positive index households are considered (infection or coinfection), the proportion of households with at least one case positive by PCR or serology was 87.9% (29/33).

No pertussis infection was detected in 7 households from 4 and 3 *B. pertussis* and *parapertussis* positive index cases respectively, representing 27 people. These households were composed of 3 [2–5] people in median [IQR], including 3 [2–3], 1 [0–1] and 0 [0–1] adults, children older than 5 years and children younger than 5 years in median [IQR], respectively.

**Possible contaminators and contaminated description.**   Among the 157 contact cases investigated, 27 (17.2%) were considered as possible contaminators, among whom 7 (26.0%) mothers and 4 (14.8%) siblings older than 5 years. Median [IQR] age was 27.3 [6.2–41.5] years, ranging from 1 to 72 years.

Sixty-seven (42.7%) contact cases were categorized as possibly contaminated, among whom 17 (25.4%) siblings older than 5 years and 12 (17.9%) aunts and uncles. The median [IQR] age was 13.2 [4.7–29.1], ranging from 3 months to 70 years. People considered as possible contaminators of index cases were significantly older than those considered as possibly contaminated by the index case (p = 0.03) (S3 Table).

Vaccination information was obtained for 39 participants. Fourteen children had received their third vaccine dose for primovaccination in the last year. Among them, we observed that three had a positive PCR for *B. pertussis*.

## Sero-epidemiological cross-sectional study

A total of 444 children and adolescents were enrolled over a period of 6 months (January to June 2021). Two children refused blood sampling and were excluded from the serology analysis. The sex ratio (M/F) was 0.82 and the median [IQR] age was 7 [5–10] years.

**Pertussis immunization history and compliance.**   Regarding primovaccination compliance, 279 (62.8%) of children were fully compliant when the strict definition of compliance is applied (S2 Fig). When we applied the less stringent definition, 365 (82.2%) children were fully compliant with the EPI primary vaccination schedule.

197 (44.4%) children did not receive any booster whilst 247 (55.6%) received a first booster at 1.6 [1.3–2.5] years in median [IQR]. Among them, 9 (2.0%) had received a second booster, representing 3% of eligible children (i.e. older than 6 years). These children received their second booster at 7.2 [6.6–7.8] years of age in median [IQR] (S4 Table).

**Pertussis serological analysis.**   Overall, 96 (21.7%) of children and adolescents presented with anti-PT IgG levels above 40 UI/mL, among whom 18 (18.8%) with a serology above 100 UI/mL.

In the population of individuals without any boosters (n = 197), 19 had received their third vaccine dose after 6 months of age and were excluded from the analysis. The proportions of children with a serology above 40 UI/mL were 20.7% (17/82), 7.6% (6/79) and 53.3% (8/15) for the age categories 3 to 5 years, 6 to 11 years and 12 to 15 years, respectively. The proportions of individuals with a positive serology were higher for the age categories 3 to 5 years (p = 0.02) and 12 to 15 years (p<0.01) compared to the 6 to 11 years category. The proportion of positive serology was also significantly higher in the 12 to 15 years category compared to 3 to 5 years (p = 0.01) (Fig 2).

In the population of individuals given one booster (n = 238), 25 had received their third vaccine dose after 6 months of age, 4 had received their first booster before 12 months of age and 77 had received their booster after 24 months of age. They were excluded from the analysis. Of the 132 remaining children, the proportions of children with a serology above 40 UI/mL were 25.6% (10/39), 25.3% (19/75) and 16.7% (3/18) in the age categories 3 to 5 years, 6 to 11 years and 12 to 15 years, respectively. No significant difference was observed between the three age categories. Among the children with positive serology in the 3 to 5 years category, 9 (23.1%) had a serology result from 40 to 100 UI/mL and one (2.6%) had a serology result above 100 UI/mL (Fig 2).

When comparing the population having received no booster with the population having received one booster, the proportion of positive serology was lower for the 6 to 11 years age

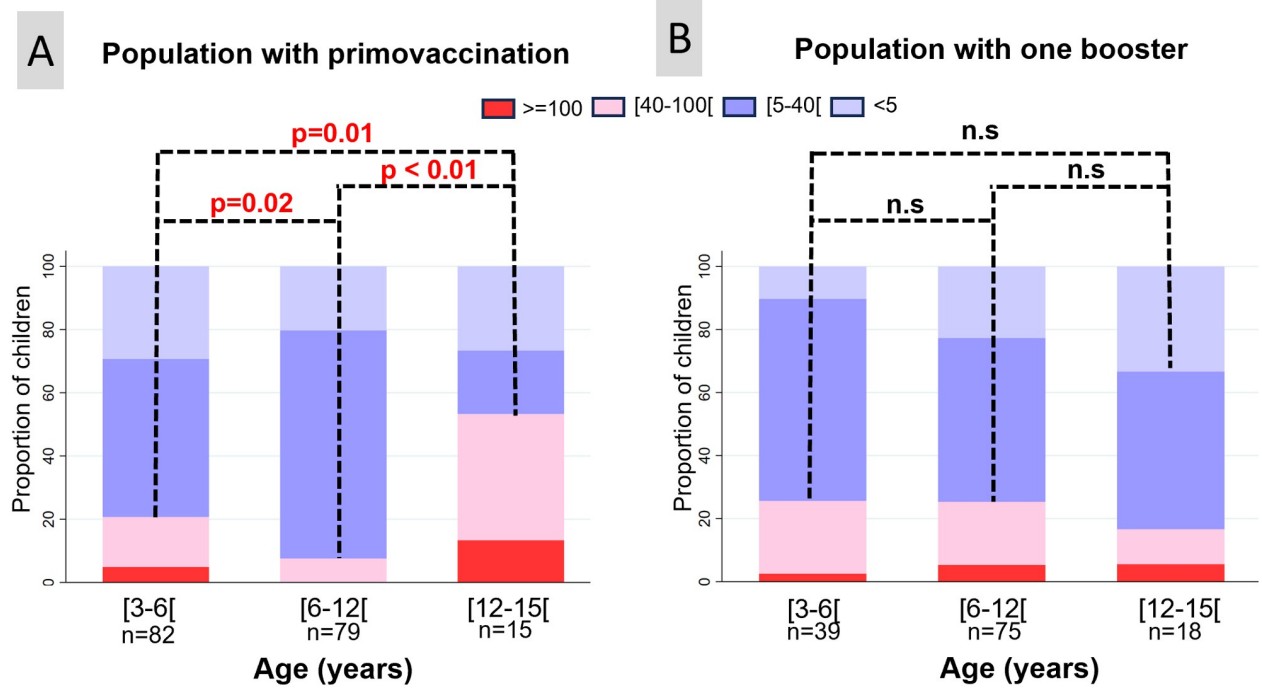

**Fig 2. Rate of seropositive individuals, computed with a threshold of 5, 40 and 100 IU/mL as a function of age, with the categories 3 to 5 years, 6 to 11 years and over 12 years.** A, left graph: among individuals who received primovaccination only; B, right graph: among individuals who received one booster.

category (p<0.01), tended to be higher for the 12 to 15 years category (p = 0.06) and was not significantly different for the 3 to 5 years category.

## Discussion

Using PCR and serology, we highlighted *B. pertussis* circulation in Abidjan, where the WHO EPI recommends three doses of wP at six, ten and fourteen weeks of age, but where surveillance of the disease is lacking. The different parts of the presented study all revealed the high circulation of *B. pertussis*.

In the hospital-based study, *B. pertussis* infection was found in 22.5% of infants below six months of age with a pertussis-like syndrome, a highly vulnerable population at risk of severe pertussis including death. While most infections were caused by *B. pertussis*, 2.1% of the cohort was infected with *B. parapertussis* alone and 4.8% of the cohort was coinfected with *B. pertussis* and *B. parapertussis*. No *B. holmesii* was identified. The presence of *B. parapertussis* observed in our study was consistent with others in the Maghreb [29–31].

Importantly, the prevalence of *B. pertussis*/*B. parapertussis* in our cohort might be slightly underestimated since three infants with a negative PCR had received macrolides at least 72 hours before inclusion, which can lead to PCR negativation. Moreover, five other infants with a negative PCR had been coughing for more than 45 days before inclusion, an interval after which PCRs may be negative.

We found 75.0% of EPI compliance among vaccine-eligible infants, i.e. of or above 6 weeks of age. This estimate, based on vaccination booklets, is stringent since it includes adherence to the vaccination schedule at six, ten and fourteen weeks in addition to vaccination coverage. In

our cohort, after careful examination of vaccination dates, only eight infants should be considered fully vaccinated (i.e. three doses) and infected. Similarly in the contact-case study, we observed three children fully vaccinated and infected by *B. pertussis/parapertussis* in the year following their third vaccine injection. wP efficacy against pertussis has repeatedly been shown to vary dramatically across vaccine producers [7, 32], with an efficacy estimated at 36% or 41% for certain previously licensed vaccines [8, 33]. Even recent wP were shown to induce various levels of immunogenicity, with seroconversion rates varying from 43.9 to 71% [9, 34]. Another factor to consider is malnutrition, which is ubiquitous in Côte d'Ivoire and could therefore contribute to the high prevalence of pertussis we observed. According to a WHO report, 20.9% of children under five were underweight in Côte d'Ivoire in 2016 and 14% were suffering from wasting [35]. In our cohort, stunting and wasting were found in 25.6% and 13.9% of the infants respectively, with stunting significantly more prevalent among infected children than in the rest of the cohort.

When considering the risk factors of pertussis infection, we showed that low BMI was a risk factor for whooping cough, consistent with the common finding that malnutrition is associated with an increased risk of infection, including respiratory tract infections [36, 37]. Moreover, respiratory infections can cause a loss of critical body stores of protein and energy [38], as observed in a Nigerian cohort where children below six months infected with pertussis lost weight and had difficulty regaining it [39]. Therefore, the association between low BMI and whooping cough could go both ways, through either malnutrition increasing the risk of infection or infection impacting weight and/or growth in these young infants. These effects could coexist as well. It has also been proposed that there could be a link between malnutrition and ineffective immunological memory mechanism [40, 41]. Malnutrition might have lowered the protective effect of vaccination in our cohort, which could explain why receiving two or three doses, versus one dose or none, does not appear to be a protective factor against infection. Other risk factors revealed by our study were age below three months and apnea. Over two thirds of the infected infants (69.1%) in our cohort were younger than three months, significantly more than among the infants older than three months despite a balanced number in the two groups. This is not surprising since, until three months of age, infants cannot be fully vaccinated. Apnea was the sole clinical sign significantly associated with pertussis infection. Other clinical symptoms such as cyanosis were not associated with infection, probably due to the symptoms' lack of specificity and the study's lack of power. The duration of cough was also not significantly associated but tended to be. The study's lack of power could explain this lack of significant association. Note that there was a significantly higher proportion of persistent cough among positive cases relative to negative cases during the follow-up consultation, which was conducted fourteen days after enrollment in the study. In addition, it was unsurprising that certain clinical signs such as inspiratory whoop were not found to be significantly associated with infection due to the recruited population's young age, as described previously [42]. Finally, we found a lower proportion of individuals infected with pertussis between July 2020 and July 2021 than for the previous inclusion period. Whooping cough undergoes cycles every three to five years, which could play a role in that effect, but this could not be assessed due to inadequate pertussis surveillance in Côte d'Ivoire. Another factor that could explain the lower ratio of pertussis infections in the sample included between July from 2020 and July 2021 might be the COVID pandemic: social distancing surely decreased the number of pertussis cases, through decreased transmission of respiratory diseases. At the same time, the number of infants with cough due to the SARS-CoV-2 (negative pertussis cases) certainly increased in hospitals and thus in our sample.

Our contact cases investigation conducted in 36 households, representing 158 contact cases with a median of 5 contacts per household, is an especially large household case contact cohort.

We highlighted the spread of infection within households, with almost half (48.7%) of contact cases infected across all age categories. Positive cases were found in 81% of the households. However, we may have underestimated the number of positive cases since serology analyses do not reveal *B. parapertussis* infections, i.e. we might have missed cases of past infections with *B. parapertussis*, which cannot be identified with PCR. The high circulation of *B. pertussis* and *B. parapertussis* within households is certainly favored by their very large size of households, for instance much larger than what was found in Tehran in a similar study [21]. Twenty-six contact cases were identified as potential contaminators and 67 as potentially contaminated by the index case. Contaminators were significantly older than contaminated individuals. Mothers formed the largest share of contaminators. Siblings and cousins older than five years formed a large share of the individuals possibly contaminated by infants. While precise vaccination status was not available for these siblings and cousins, the high contamination rate among them suggests significant susceptibility to pertussis in that age group, consistent with our sero-epidemiology study's findings.

In the sero-epidemiological study, the population of fully primovaccinated individuals was compliant with the EPI vaccination schedule (82.2%), similar to compliance previously described in Togo and Madagascar [18]. Regarding boosters, which are recommended nationwide at sixteen months, six and a half years and eleven and a half years of age, a low (55.6%) and extremely low (2%) proportion of individuals had received one or two boosters respectively. Financial barriers explain this poor access to booster vaccines.

Taking only individuals who received their last injections on time, high levels of anti-PT antibodies were observed in the youngest age category, three to five years, in both the booster and non-booster groups, reflecting the fact that infections can occur even within 4–5 years of the last vaccine dose. Another explanation for the relatively high rate of seropositivity in the younger age group with one booster could be possible booster-induced IgG, leading to an overestimation of infections in this group. However, with the 40 IU/mL threshold applied, the proportion should be very small.

In the group that received primovaccination only, contact with the bacterium in the years after the last injection is consistent with lower levels of anti-PT antibodies in the group aged six to eleven (natural booster acquired during the previous period and therefore less susceptibility to the disease) and with an increase in the group older than twelve years (disease susceptibility restored after the previous period due to waning immunity, even after infection).

In the group that received one booster, the significantly higher rate of seropositivity among children aged six to eleven years relative to the non-booster group could be interpreted as the consequence of lower levels of infection in younger age ranges, due to the booster's protective effect. The opposite trend for the next age range could be interpreted as less susceptibility to the disease due to the natural booster acquired previously. However, given our study's lack of power and possible early natural boosters, we were unable to observe a significant difference between age groups.

## Limitations and strengths

Our study is characterized by the richness of the information collected: vaccination history (number of doses and vaccination dates) was systematically collected in the PS and CCES, and pertussis infections, past or present, were detected with reliable laboratory methods (serology and PCR) on the three populations studied. In hospitals, retrospective and prospective clinical data on infants at two timepoints were collected, as well as information on antibiotic uptake and biological exams, whenever available. As for the quality of the data, we devoted significant resources to the training of the local teams of clinicians and biologists taking part in the study,

and opted for multivariate analyses to control for biases arising from the richness of the dataset. Our study therefore provides new data on the recent epidemiology of pertussis in Abidjan in various age ranges, and a real-life snapshot of the management of pertussis in Abidjan hospitals, while the contact-cases investigation offers an insight into the spread of the contamination through the large size of the cohort. This global and detailed picture of the situation could inform policy-maker decisions.

The study limitations relate mainly to a lack of statistical power. Our study's lack of power could lead to the failure to identify some risk factors, such as duration of cough. Moreover, we did not investigate the presence of pathogens other than *Bordetella* species, which might have obscured the investigation of whooping cough symptoms. In the PS, three infants had received macrolides before the study, which are known to negativate PCR, and five infants had been coughing for more than thirty days while PCR is sensitive only during the first three weeks of symptoms. We might therefore have underestimated pertussis prevalence.

In the contact-cases investigation, we may have lacked the sensitivity to detect contaminators and contaminated individuals for a number of reasons, such as underreported cough or mistakes relating to recall biases for the self-reporting of cough duration, or long delays in including a contaminated contact case leading to a negative PCR.

In the sero-epidemiological study, our conclusions are limited by a lack of power.

## Conclusions

In the three populations studied in real-life conditions in Abidjan, our data reveal high pertussis circulation, which makes infants who are too young to be fully vaccinated especially at risk of pertussis infections and thus at risk of severe, sometimes fatal forms of the disease. The high level of circulation of the bacteria in the population of children under 6 months of age (PS), and in particular under 3 months of age (not yet fully primo-vaccinated), reinforces recommendations for vaccination around pregnant women and during pregnancy. Primary vaccination and booster administration in Abidjan is critical: three primovaccination doses on time protects infants from infections or at least attenuates severe forms of the disease. In the PS, a relatively high proportion of the infants appeared to be compliant with the EPI schedule. Furthermore, vaccination history in the SECS also revealed that only half of the children and adolescents recruited had received boosters. The high level of circulation of pertussis observed among these children and adolescents (SECS) indicates that booster should be broadly administered to interrupt transmission chains to protect infants who are too young to be fully vaccinated. However, vaccine boosters are out-of-pocket expenses in Côte d'Ivoire, contrary to primary doses. Barriers to implementing the boosters recommended by the WHO and national vaccine schedule, especially financial accessibility, need to be overcome. The vaccination of pregnant women and healthcare workers to stop severe disease and deaths below three months of age could also be considered.

Our study also revealed low BMI as a risk factor for the disease, suggesting that children's nutritional status must absolutely be considered to ensure more effective disease control in general, and the control of vaccine-preventable diseases in particular.

Finally, our study introduced laboratory capacity, and training to PCR/serology procedures/clinical detection of the disease, which could have key impacts at individual and collective levels. Indeed, detection of the disease, especially at early stages, enables prompt action during the catarrhal period (antibiotic therapy, respiratory isolation, vaccination of surrounding populations) to reduce symptoms for individuals and transmission for surrounding populations. In addition, an effective national surveillance system, with laboratory capacity and training, enables vaccine strategies to be tailored to the population profile and evaluated.

## Supporting information

**S1 File. Alternative language abstract, in French.**
(TIF)

**S1 Fig. Decision tree flowchart for *Bordetella* species identification for biological diagnosis using qPCR assays.**
(TIF)

**S2 Fig. Description of compliance for each primary vaccination dose based on the age at injection recommended by the Expanded Program on Immunization (EPI) [12] and pertussis national vaccination schedule [11], i.e. six, ten and fourteen weeks of age.** Acceptable age intervals where infants are considered compliant are indicated in the green and blue boxes. The intervals around recommended injection age where the dose is considered as received too early are indicated in yellow boxes, and the intervals where the dose is considered as received too late are indicated in red.
(TIF)

**S1 Table. Biological exam results.**
(TIF)

**S2 Table. Population description of all contact cases within PCR positive cases (*B. Parapertussis* infections, *B. pertussis* infections, coinfections), as well as within serology positive cases with the 40 UI/mL threshold (*B. pertussis* infections).**
(TIF)

**S3 Table. Description of the population with serological and PCR analyses as possible contaminators versus possibly contaminated, whenever possible.** n = 157 since missing serology and PCR data for one individual.
(TIF)

**S4 Table. Description of vaccination ages in the sero-epidemiological study for the primo-vaccination (n = 444) and boosters (n = 247 with one booster; n = 9 with two boosters).**
(TIF)

**S5 Table. Comparison of clinical signs between infected infants having received either 0 or 1 vaccine injection (n = 33) or 2 or 3 injections (n = 9).**
(TIF)

## Acknowledgments

We thank all the children and their parents or legal guardians for participating in this study. We thank Dr Gilbernaire Elia for his help in facilitating contacts to conduct the study, the Directorate General of Health, the heads of the Centre Hospitalier et Universitaire d'Angré, of the Hôpitaux Généraux de Yopougon Attié and of Port-Bouët and of the Centre de Santé Urbain à base Communautaire d'Angré. We also thank the pediatricians and nurses working in the department of pediatry in the Centre Hospitalier et Universitaire d'Angré, des Hôpitaux Généraux de Yopougon Attié et de Port-Bouët et du Centre de Santé Urbain à base Communautaire d'Angré. We thank the Abidjan 1 Regional Directorate of the Ministry of National Education. We also wish to thank the Directorate of Mutuality and Social Work in Schools (DMOSS), the Preschool and Primary Education Inspections of Cocody-Blockhauss and Cocody-Akouedo, the principals and staff of participating schools, the Technical and Technological Department (DTT), the Transport and Packaging Unit (UTEP) and the Supply

Management Unit (UGA) of the Institut Pasteur de Cote d'Ivoire for logistical support. Finally, we wish to thank the Côte d'Ivoire PERILIC team, at the Institut Pasteur of Côte d'Ivoire. We also thank the Clara Belliveau Foundation for its contribution in gaining scientific knowledge on pertussis epidemiology and disease.

## Author Contributions

**Conceptualization:** Man Koumba Soumahoro, Mohand Ait Ahmed, Marie-Pierre Ouattara-Abina, Nicole Guiso, Fabien Taieb.

**Data curation:** Man Koumba Soumahoro, Gaëlle Noel, Mahdi Rajabizadeh.

**Formal analysis:** Mahdi Rajabizadeh, Fabien Taieb.

**Funding acquisition:** Man Koumba Soumahoro, Gaëlle Noel, Denis Macina, Nicole Guiso, Fabien Taieb.

**Investigation:** Man Koumba Soumahoro, Constant Joseph Koné, Daouda Sévédé, Kan Stéphane Kouassi, Yves Landry Kangah, Mariam Deme-Dramé, Kouamé Mathias N'Dri, Marie-Pierre Ouattara-Abina, Narcisse Tano, Fadima Sylla, Melissa Cardenat.

**Methodology:** Man Koumba Soumahoro, Fabien Taieb.

**Project administration:** Man Koumba Soumahoro, Gaëlle Noel, Mohand Ait Ahmed.

**Resources:** Man Koumba Soumahoro, Fabien Taieb.

**Software:** Gaëlle Noel, Mohand Ait Ahmed, Fabien Taieb.

**Supervision:** Man Koumba Soumahoro, Gaëlle Noel, Nicole Guiso, Fabien Taieb.

**Validation:** Man Koumba Soumahoro, Nicole Guiso, Fabien Taieb.

**Visualization:** Florence Campana, Mahdi Rajabizadeh.

**Writing – original draft:** Florence Campana, Nicole Guiso, Fabien Taieb.

**Writing – review & editing:** Man Koumba Soumahoro, Constant Joseph Koné, Mahdi Rajabizadeh, Daouda Sévédé, Mohand Ait Ahmed, Kan Stéphane Kouassi, Yves Landry Kangah, Mariam Deme-Dramé, Kouamé Mathias N'Dri, Marie-Pierre Ouattara-Abina, Narcisse Tano, Fadima Sylla, Melissa Cardenat, Denis Macina.

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
