## [Decision Letter · Decision Letter 0]

17 May 2024

PONE-D-24-10827High pertussis circulation among infants, children and adolescents in Abidjan, Côte d’IvoirePLOS ONE

Dear Dr. Taieb,

Thank you for submitting your manuscript to PLOS ONE. After careful consideration, we feel that it has merit but does not fully meet PLOS ONE’s publication criteria as it currently stands. Therefore, we invite you to submit a revised version of the manuscript that addresses all the points raised during the review process. The conclusion should be revised to more accurately reflect the data presented.

We look forward to receiving your revised manuscript.

Kind regards,

Daniela Flavia Hozbor

Academic Editor

PLOS ONE

Journal Requirements:

"I have read the journal's policy and the authors of this manuscript have the following competing interests : Denis Macina is an employee of Sanofi and declares holding shares of the Sanofi group as part of his compensation. The other authors declare that no competing interests exist."

Reviewers' comments:

Reviewer's Responses to Questions

**Comments to the Author**

1. Is the manuscript technically sound, and do the data support the conclusions?

Reviewer #1: Partly

Reviewer #2: Yes

2. Has the statistical analysis been performed appropriately and rigorously? 

Reviewer #1: Yes

Reviewer #2: Yes

3. Have the authors made all data underlying the findings in their manuscript fully available?

Reviewer #1: No

Reviewer #2: Yes

4. Is the manuscript presented in an intelligible fashion and written in standard English?

Reviewer #1: Yes

Reviewer #2: Yes

5. Review Comments to the Author

Reviewer #1: The study investigated pertussis infection in a country with limited surveillance system of the disease using various method and found high circulation of the disease. The study also investigated vaccination compliance among a portion of the study population.

Here are few comments for considerations:

1. The use of cohort for the hospital sample is problematic. This is not a cohort study; I suggest that the others use the term hospital-based sample instead of cohort study. In fact, it can be considered like a cross sectional study.

2. In the abstract, please spell out CCIS (line 53)

3. Line 62 in the abstract and later: Provide a clear definition of “possibly contaminated” how did you determined who is contaminant and who is possibly contaminated?

4. Lines 155-167: After test was repeated if there are still inconsistency what did you do with that sample. How many samples were in such situation?

5. Line 202: Epidemiological case definition is a weak definition because a contact could have infection without transmission to the infants. Suggest removing this from the definition.

6. Although the study was not to evaluate vaccine effectiveness, it looks like you had all the ingredient to have a crude estimate of the VE.

In this study based on your vaccine compliance results and the number of people who were infected it looks like the VE was initially high. It would be important to comment on whether the overall high pertussis infection seen is due to waning or to under vaccination. It seems to me that 75% compliance is good and yet you still see high infection proportion.

7. The study has information on clinical presentation of vaccinated and unvaccinated cases, it would be important to describe the clinical presentation of the breakthroughs and the unvaccinated.

8. Table 2 shows that antibiotic before inclusion is associated with pertussis infection. I think this need to be further explained so that it is not misinterpreted as a risk factor. What is the reason for not showing the Ora for the other variables in the table? What is CV in DTP, please spell out in footnote.

9. Lines 531 – 533: The sentence is confusing. I would imagine that the barrier measures and social distancing introduced due to the pandemic would decrease cough due to respiratory diseases instead of increase.

10. The study enumerated the limitations but are there any strengths?

11. Conclusion: The conclusion is very broad with another of recommendations that are not always supported by the study results. I suggest revising the conclusion to reflect the study’s results.

Reviewer #2: This manuscript by Soumahoro et al. investigates the circulation of pertussis and household-related cases within an African cohort. Africa reports the highest proportion of total pertussis cases (33%) and the highest proportion of worldwide pertussis-associated deaths (58%). These numbers are likely underestimations due to underdiagnosed and underreported cases. Between April 2019 and July 2021, 187 infants were enrolled. Among them, 22% tested positive for pertussis, with the majority caused by Bordetella pertussis (BP), followed by Bordetella parapertussis (BPP), and coinfections of BP and BPP. Strikingly, 81% of positive cases were detected in the context of co-housing.

There are several points that require further clarification and expansion:

Vaccine Schedule: In line 90, where the vaccine schedule is detailed, it is important to specify which vaccine is used as well as differences with other countries.

References: Important statements lack references, such as, “High vaccine coverage and compliance with this three-dose schedule for infants in their first year of life is not sufficient for B. pertussis to be controlled” (lines 94-96). This is an example, the references in general need to be revised.

Sample Storage: The manuscript states that nasopharyngeal swab (NPS) samples were stored at 4°C until DNA extraction was performed within two days. However, it also mentions that when immediate extraction was not possible, samples were stored at -20°C. Clarification is needed on the storage conditions. Were cryoprotectants used to ensure the integrity of the samples? Did the authors perform a validation to confirm that freezing did not affect the results?

Clarification on DNA Extraction: In line 169, it is stated, “Total DNA was extracted from 100 μl of the transport media (pure sample) and from 10 μl of the transport media spiked into 90 μl of sterile water (diluted sample) using the commercial High Pure PCR Template Preparation Kit (Roche). Please revise the language, it is confusing and the explanation as to why two different extractions were done should be at the beginning.

Line 172: “Were tested for possible qPCR inhibition”. This needs to be clarify, what do the authors mean by PCR inhibition?

Rigor and Reproducibility: The materials and methods section should include more references and detailed specifications to ensure rigor and reproducibility.

Table 1: Needs more description on how the p-values were calculated. But in general figure legends need to be revised to guarantee statistical test are added.

Table 2: The meaning of “ref” should be clarified.

Overall, this is a good manuscript of interest for the scientific community.

6. PLOS authors have the option to publish the peer review history of their article (what does this mean?). If published, this will include your full peer review and any attached files.

Reviewer #1: No

Reviewer #2: No

---

## [Author Response · Author response to Decision Letter 0]

29 Jul 2024

Reviews from the editors and answers

Here are few comments for considerations:

1. The use of cohort for the hospital sample is problematic. This is not a cohort study; I suggest that the others use the term hospital-based sample instead of cohort study. In fact, it can be considered like a cross sectional study.

We thank the reviewer for her/his comment. We would like to specify that each infant in the hospital sample was seen twice: (i) at inclusion, where clinical presentation, antibiotic treatment were assessed and NPS collected, and (ii) during follow-up consultation (median [IQR] of 13 [13-14] days, from 13 to 21 days), where the disease outcome was assessed. We therefore suggest to change by “Hospital-based prospective study” in order to be also in coherence with the previous article that we published on the same study in Iran {Noel, 2021 #330}.

We implemented this change in the manuscript and replaced CS for “Cohort study” with PS throughout the manuscript.

2. In the abstract, please spell out CCIS (line 53)

We thank the reviewer for this observation, and replaced CCIS with CCS, since CCIS was referring to ‘contact-cases investigation’ (CCS).

3. Line 62 in the abstract and later: Provide a clear definition of “possibly contaminated” how did you determined who is contaminant and who is possibly contaminated?

We used the combination of information from the qPCR results, the serological analyses and the relative start of cough of the index and contact cases to determine, when possible, whether contact cases were more likely to be contaminators or contaminated.

Typically, when a contact case cough had started before the index cough, and presented with a positive serology, then this person was considered as a possible contaminator. When a contact case had started coughing several days after the index and had a positive qPCR, then this person was considered as a possibly contaminated person – since this person was infected but with a delay relatively to the index case.

We refer to ‘possible contaminator’ and ‘possibly contaminated’ to make it clear that the categorization is a hypothesis based on the information available (symptoms, qPCR and serological results, as well as based on the fact that the positive index and contacts had been in close contact for at least five consecutive days before inclusion of the index case). 

We detailed this definition lines 218-224.

4. Lines 155-167: After test was repeated if there are still inconsistency what did you do with that sample. How many samples were in such situation?

We would have excluded participants if inconsistencies had been remaining but, fortunately, after test repetition, results were interpretable and all the samples could be kept. We rewrote this section (lines 159-162) of the manuscript, that could lead to confusion and thank the reviewer for their careful reading.

5. Line 209: Epidemiological case definition is a weak definition because a contact could have infection without transmission to the infants. Suggest removing this from the definition.

Thank you for this comment, which gives us the opportunity to clarify in the manuscript (line 211) the epidemiological confirmation procedure we used: we considered the contact cases for IS481+ only index case, i.e. for individuals infected with a Bordetella species only, but whose strain was not revealed with the ptxP and IS1002. We considered these patients as positive cases (for epidemiological reason) in case of Bp identification in at least one contact case. We considered these patients as negative in case of absence of positive contact case for Bp.

The epidemiological link definition retained here is stricter than that used in the case definitions of France, eCDC, Canada or Massachusetts (Cherry, CID 2012).

6. Although the study was not to evaluate vaccine effectiveness, it looks like you had all the ingredient to have a crude estimate of the VE.

In this study based on your vaccine compliance results and the number of people who were infected it looks like the VE was initially high. It would be important to comment on whether the overall high pertussis infection seen is due to waning or to under vaccination. It seems to me that 75% compliance is good and yet you still see high infection proportion.

Thank you for this interesting comment. The study does unfortunately not enable us to assess crude VE, since our sample includes only infants with symptoms-we thus miss for example all the infants vaccinated, exposed to Bp and without symptoms. Moreover, we can’t estimate the exposition to the pathogen, and we didn’t recruit control groups regarding the number of vaccine injections received.

Due to the young age of the cohort, we defined compliance as follows: “the number of doses received by an individual relative to the number of doses expected at their age, among eligible children (i.e. older than 6 weeks)”, lines 254-259, implying that compliance is independent from full primovaccination. Furthermore, in the definition, upper limit of age to receive the first vaccine injection was before 10 weeks of life, meaning that children between 6 and 10 weeks of age without vaccine injection weren’t considered as not compliant. Estimation of the compliance was to describe the state of the application of EPI in Ivory Coast and not to estimate vaccine effectiveness. 

Therefore, 75% compliance in the cohort shows that the EPI recommendations were largely followed in this population. However, only 50.3% of the cohort had received two or three doses, since half of the cohort was under three months of age i.e. too young to be fully vaccinated.

However, it is interesting to note that among the infants having received three doses and thus fully primovaccinated, 8 out of 41 were sick (19.5%). In regard of this high proportion, noteworthy (as described in the manuscript), low BMI was a risk factor in this cohort. Possibly, the nutritional status played a role in the high levels of infection, including among infants fully vaccinated, through an alteration of the immune response. 

The table below suggests that stunting among the infants fully vaccinated might have been more important among infected infants relatively to non-infected infants (cf Table 1, in trend). We also consider prematurity in this group as a possible factor associated with infections, but none of the infected infants fully primovaccinated was born prematurely.

In other words: the high levels of infection in the cohort does not speak much about vaccine protection. Two thirds of the infected infants (69.1%) in our cohort were younger than three months while they represent half of the cohort, which highlights the vulnerability of infants under three months, too young to be fully vaccinated, and reveals the high level of circulation of pertussis in Abidjan among people in the vicinity of infants. Other factors might be at play in the high infection rate, such as nutritional deficiencies. All these points were in line with cocooning strategy and vaccine injection during pregnancy recommendations. 

Table 1. Comparison of the BMI, Wasting, Stunting rates in the population of infants having received three vaccine doses, depending on their clinical status (infected or not). 

7. The study has information on clinical presentation of vaccinated and unvaccinated cases, it would be important to describe the clinical presentation of the breakthroughs and the unvaccinated.

Symptoms caused by pertussis infections differ by age notably between the first three months of life and later (Cherry et al., 2012). Symptoms must therefore be compared within relevant age categories (under or above three months). 

 We report below the clinical symptoms among infants older than three months to investigate the impact of injections on symptoms, but it is worth noting that statistical power is lacking, and prevents firms conclusions to be drawn. 

We only consider the number of injections administered at least 14 days before symptom onset. There was no difference in the clinical presentation at inclusion of positive cases above three months with zero or one dose (n=4), vs. two or three doses (n=9), but this analysis lacked power (cf Table 1 below). The positivity rate was 20% (4/20) among the infants having received no dose or one dose vs. 16% (9/55) among the infants having received two or three doses (all older than three months). 

We ran the same analysis with no age threshold, in order to have more statistical power (cf Table 2), granted that this could introduce a bias due to the association between the number of doses and the age, but this analysis did not reveal significant differences either 

Table 2. Comparison of the symptoms in infants having received maximally one injection, vs. two or three injections. Upper part: within infants older than three months. Lower part: all the infants.

Moreover, the design and objectives of the study didn’t allow to determine the impact of vaccination on the severity of the disease.

For all these reasons, we didn’t present these results in this manuscript already containing a high number of information and analysis. 

8. Table 2 shows that antibiotic before inclusion is associated with pertussis infection. I think this need to be further explained so that it is not misinterpreted as a risk factor. What is the reason for not showing the Ora for the other variables in the table? What is CV in DTP, please spell out in footnote.

We thank the reviewer for this important point. In the table 1 of the manuscript, antibiotherapy is associated with pertussis infection (univariate analysis), but this analysis does not control for correlations between factors. In fact, exposure to antibiotics was associated with the number of consultations, itself associated with cough duration, and with apnea. Taking into account the different confounding biases (multivariate analysis), we showed that apnea was a risk factor for pertussis, and not prior antibiotic exposure. 

Ora are not reported when not significant for clarity of presentation and as commonly accepted.

DTP-CV (containing vaccine) is indicated lines 316 and 372 in footnote.

9. Lines 531 – 533: The sentence is confusing. I would imagine that the barrier measures and social distancing introduced due to the pandemic would decrease cough due to respiratory diseases instead of increase.

The phrasing was confusing indeed, we reformulated the sentences (line 548 to 552). Noteworthy, the variable semester of inclusion was included in the multivariate analysis not as a possible risk factor but to control for all the possible biases, including seasonality. 

10. The study enumerated the limitations but are there any strengths?

We indeed added a paragraph (lines 595-607) on the strengths of the study (namely the quality and richness of the information gathered), and thank the reviewer for this suggestions.

11. Conclusion: The conclusion is very broad with another of recommendations that are not always supported by the study results. I suggest revising the conclusion to reflect the study’s results.

We indeed revised the conclusion to make explicit the links between the conclusion and the results of the study.

Reviewer #2: This manuscript by Soumahoro et al. investigates the circulation of pertussis and household-related cases within an African cohort. Africa reports the highest proportion of total pertussis cases (33%) and the highest proportion of worldwide pertussis-associated deaths (58%). These numbers are likely underestimations due to underdiagnosed and underreported cases. Between April 2019 and July 2021, 187 infants were enrolled. Among them, 22% tested positive for pertussis, with the majority caused by Bordetella pertussis (BP), followed by Bordetella parapertussis (BPP), and coinfections of BP and BPP. Strikingly, 81% of positive cases were detected in the context of co-housing.

There are several points that require further clarification and expansion:

Vaccine Schedule: In line 90, where the vaccine schedule is detailed, it is important to specify which vaccine is used as well as differences with other countries.

We thank the reviewer for her/his comment. We edited the manuscript to address this interesting remark concerning the vaccine used in the PEV in Côte d’Ivoire, and the contrast with high-income countries, where acellular vaccines are available for primary vaccination (lines 93-98).

References: Important statements lack references, such as, “High vaccine coverage and compliance with this three-dose schedule for infants in their first year of life is not sufficient for B. pertussis to be controlled” (lines 94-96). This is an example, the references in general need to be revised.

We added the reference (Zepp, Heininger et al. 2011) (line 102) regarding the fact that compliant primovaccination is not sufficient for B.pertussis to be controlled.

Boosters are needed since, as shown in {Wendelboe, 2007 #243}, vaccination against pertussis does not provide lifelong protection (and neither infection does). Therefore, boosters are key to prevent older age groups, infected after the waning of vaccine-induced protection, from contaminating infants who are at risk of developing severe, sometimes fatal, forms of the disease and who are too young to be fully vaccinated.

Several studies have indeed shown, in Europe and in the US, that there is a shift of infections toward children, adolescents and adults, who have been contributing to an increase in the incidence in non-vaccinated (or not fully vaccinated) infants (Zepp, Heininger et al. 2011).

We also added a reference on the reactogenicity of cellular vaccines (Kitchin, Southern et al. 2006).

Sample Storage: The manuscript states that nasopharyngeal swab (NPS) samples were stored at 4°C until DNA extraction was performed within two days. However, it also mentions that when immediate extraction was not possible, samples were stored at -20°C. Clarification is needed on the storage conditions. Were cryoprotectants used to ensure the integrity of the samples? Did the authors perform a validation to confirm that freezing did not affect the results?

We thank the reviewer for these questions, which enabled us to edit the manuscript. Actually, samples were frozen at -80 degrees Celsius when DNA extraction was not performed within 48 hours (this was corrected in the manuscript, instead of -20 degrees). It has been shown that storage at -20°C or -70°C protects DNA from degradation (Kim, Choi et al. 2012) (reference added) ; we therefore did not use cryoprotectants nor did we run tests about DNA integrity. We chose a temperature of -80°C.

Clarification on DNA Extraction: In line 169, it is stated, “Total DNA was extracted from 100 μl of the transport media (pure sample) and from 10 μl of the transport media spiked into 90 μl of sterile water (diluted sample) using the commercial High Pure PCR Template Preparation Kit (Roche). Please revise the language, it is confusing and the explanation as to why two different extractions were done should be at the beginning.

Line 172: “Were tested for possible qPCR inhibition”. This needs to be clarify, what do the authors mean by PCR inhibition?

We rephrased the sentence (lines 173-180) to clarify that it consists of one extraction with a dilution procedure, to address the issue of too large quantity of DNA inhibiting the qPCR that could lead to false negatives (qPCR inhibition).

Rigor and Reproducibility: The materials and methods section should include more references and detailed specifications to ensure rigor and reproducibility.

This comment enabled us to indeed increase the reproducibility of the methods through addition of references about the material used. We hope this satisfies the reviewer.

Table 1: Needs more description on how the p-values were calculated. But in general figure legends need to be revised to guarantee statistical test are added.

Thank you for this useful remark, we modified the Material and Methods section to describe the conditions in which each test is used (lines 239-241).

Table 2: The meaning of “ref” should be clarified. 

The meaning (the reference used for comparisons) is now indicated as a footnote below the Table 2.

Overall, this is a good manuscrip

---

## [Decision Letter · Decision Letter 1]

28 Aug 2024

PONE-D-24-10827R1High pertussis circulation among infants, children and adolescents in Abidjan, Côte d’IvoirePLOS ONE

Dear Dr. Taieb,

Thank you for submitting your manuscript to PLOS ONE. After careful consideration, we feel that it has merit but does not fully meet PLOS ONE’s publication criteria as it currently stands. Therefore, we invite you to submit a revised version of the manuscript that addresses the points raised during the review process.

We look forward to receiving your revised manuscript.

Kind regards,

Daniela Flavia Hozbor

Academic Editor

PLOS ONE

**Journal Requirements:**

Reviewers' comments:

Reviewer's Responses to Questions

**Comments to the Author**

1. If the authors have adequately addressed your comments raised in a previous round of review and you feel that this manuscript is now acceptable for publication, you may indicate that here to bypass the “Comments to the Author” section, enter your conflict of interest statement in the “Confidential to Editor” section, and submit your "Accept" recommendation.

Reviewer #1: All comments have been addressed

2. Is the manuscript technically sound, and do the data support the conclusions?

Reviewer #1: Partly

3. Has the statistical analysis been performed appropriately and rigorously? 

Reviewer #1: Yes

4. Have the authors made all data underlying the findings in their manuscript fully available?

Reviewer #1: Yes

5. Is the manuscript presented in an intelligible fashion and written in standard English?

Reviewer #1: Yes

6. Review Comments to the Author

**Reviewer #1:** Thank you for addressing my questions and comments.

Please add under method that the hospital sample was followed and specify that follow up was to evaluate the course of disease.

I suggest that the authors add the Table 2 in the response to supplemental materials and change the “oui” to Yes and the “non” to “No".

Please change limitation to "Limitations and strengths"

7. PLOS authors have the option to publish the peer review history of their article (what does this mean?). If published, this will include your full peer review and any attached files.

Reviewer #1: No

---

## [Author Response · Author response to Decision Letter 1]

7 Sep 2024

Please add under method that the hospital sample was followed and specify that follow up was to evaluate the course of disease. 

Thank you. We added the sentence ‘a prospective study in which infants were followed up for an assessment of the evolution of the disease. (Line 127-128). 

 I suggest that the authors add the Table 2 in the response to supplemental materials and change the “oui” to Yes and the “non” to “No". 

We indeed added the Table after replacement of the ‘Oui’/’Non’, and we mentioned this analysis (Line 336-339), as well as the Table Supplementary 5, whose caption was added at the end of the document (Line 675-676). 

Please change limitation to "Limitations and strengths" 

We added it (Line 594).

---

## [Editor Report · Decision Letter 2]

11 Sep 2024

High pertussis circulation among infants, children and adolescents in Abidjan, Côte d’Ivoire

PONE-D-24-10827R2

Dear Dr. Fabien Taieb,

We’re pleased to inform you that your manuscript has been judged scientifically suitable for publication and will be formally accepted for publication once it meets all outstanding technical requirements.

Kind regards,

Daniela Flavia Hozbor

Academic Editor

PLOS ONE
---

## [Editor Report · Acceptance letter]

14 Oct 2024

PONE-D-24-10827R2 

PLOS ONE

Dear Dr. Taieb, 

I'm pleased to inform you that your manuscript has been deemed suitable for publication in PLOS ONE. Congratulations! Your manuscript is now being handed over to our production team.

Kind regards, 

on behalf of

Dr. Daniela Flavia Hozbor 

Academic Editor

PLOS ONE